# Detection and Molecular Diversity of *Cryptosporidium* spp. and *Giardia duodenalis* in the Endangered Iberian Lynx (*Lynx pardinus*), Spain

**DOI:** 10.3390/ani14020340

**Published:** 2024-01-22

**Authors:** Pablo Matas-Méndez, Gabriel Ávalos, Javier Caballero-Gómez, Alejandro Dashti, Sabrina Castro-Scholten, Débora Jiménez-Martín, David González-Barrio, Gemma J. Muñoz-de-Mier, Begoña Bailo, David Cano-Terriza, Marta Mateo, Fernando Nájera, Lihua Xiao, Pamela C. Köster, Ignacio García-Bocanegra, David Carmena

**Affiliations:** 1Faculty of Veterinary, Alfonso X El Sabio University (UAX), 28691 Villanueva de la Cañada, Spain; pablomatas1990@hotmail.com; 2Parasitology Reference and Research Laboratory, Spanish National Centre for Microbiology, Health Institute Carlos III, 28220 Majadahonda, Spain; avalos196401@gmail.com (G.Á.); dashti.alejandro@gmail.com (A.D.); david.gonzalezb@isciii.es (D.G.-B.); begobb@isciii.es (B.B.); dacarmena@isciii.es (D.C.); 3Department of Animal Health, Animal Health and Zoonosis Research Group (GISAZ), UIC Zoonoses and Emerging Diseases (ENZOEM), University of Córdoba, 14014 Córdoba, Spain; sabrina1996cs@gmail.com (S.C.-S.); debora.djm@gmail.com (D.J.-M.); v82cated@uco.es (D.C.-T.); v62garbo@uco.es (I.G.-B.); 4Infectious Diseases Unit, Maimonides Institute for Biomedical Research (IMIBIC), University Hospital Reina Sofía, University of Córdoba, 14004 Córdoba, Spain; 5CIBERINFEC, ISCIII—CIBER Infectious Diseases, Health Institute Carlos III, 28029 Madrid, Spain; 6Faculty of Health Sciences, Alfonso X El Sabio University (UAX), 28691 Villanueva de la Cañada, Spain; gmunodem@uax.es; 7Department of Microbiology and Parasitology, Faculty of Pharmacy, Complutense University of Madrid, 28040 Madrid, Spain; mmateo14@ucm.es; 8Karen C. Drayer Wildlife Health Center, School of Veterinary Medicine, University of California, Davis, CA 95616, USA; fnajera@ucdavis.edu; 9College of Veterinary Medicine, South China Agricultural University, Guangzhou 510642, China; lxiao1961@gmail.com; 10Faculty of Medicine, Alfonso X El Sabio University (UAX), 28691 Villanueva de la Cañada, Spain

**Keywords:** epidemiology, transmission, public health, zoonosis, PCR, genotyping, *ssu* rRNA, *gdh*, *bg*, *tpi*

## Abstract

**Simple Summary:**

The Iberian lynx is an iconic feline species endemic to the Iberian Peninsula. Since the second half of the past century, its global population has decreased dramatically to the brink of extinction as a consequence of human-driven activities (habitat reduction and transformation, illegal hunting, road kills, density decrease in natural preys) and infectious diseases. Fortunately, the successful implementation of conservation programs has reversed this gloomy trend, allowing for an increase in the Iberian lynx population to over 1600 free-ranging animals in 2022. Regarding infectious diseases, very little is known on the epidemiology and health impact of the diarrhoea-causing intestinal protozoan parasites *Cryptosporidium* and *Giardia* in the Iberian lynx. To tackle these questions, we investigated the presence and molecular diversity of both pathogens in 256 collected faecal samples from 251 free-ranging and captive Iberian lynxes in Spain during the period 2017–2023. Our results demonstrate that *Cryptosporidium* (2.4%) and *Giardia* (27.9%) are present at different frequencies in the surveyed individuals. Our molecular analyses also indicate that a significant proportion of the *Cryptosporidium* infections detected are caused by strains that are typically found in the preys the Iberian lynx feed on. Interestingly, we also found that the Iberian lynx can harbour genetic variants of *Cryptosporidium* and *Giardia* with the potential to infect humans, although the likelihood of such events is judged low due to the light infections detected in the investigated animals.

**Abstract:**

*Cryptosporidium* spp. and *Giardia duodenalis* are the main non-viral causes of diarrhoea in humans and domestic animals globally. Comparatively, much less information is currently available in free-ranging carnivore species in general and in the endangered Iberian lynx (*Lynx pardinus*) in particular. *Cryptosporidium* spp. and *G. duodenalis* were investigated with molecular (PCR and Sanger sequencing) methods in individual faecal DNA samples of free-ranging and captive Iberian lynxes from the main population nuclei in Spain. Overall, *Cryptosporidium* spp. and *G. duodenalis* were detected in 2.4% (6/251) and 27.9% (70/251) of the animals examined, respectively. Positive animals to at least one of them were detected in each of the analysed population nuclei. The analysis of partial *ssu* rRNA gene sequences revealed the presence of rodent-adapted *C. alticolis* (*n* = 1) and *C. occultus* (*n* = 1), leporid-adapted *C. cuniculus* (*n* = 2), and zoonotic *C. parvum* (*n* = 2) within *Cryptosporidium*, and zoonotic assemblages A (*n* = 5) and B (*n* = 3) within *G. duodenalis*. Subgenotyping analyses allowed for the identification of genotype VaA19 in *C. cuniculus* (*gp60* locus) and sub-assemblages AI and BIII/BIV in *G. duodenalis* (*gdh*, *bg*, and *tpi* loci). This study represents the first molecular description of *Cryptosporidium* spp. and *G. duodenalis* in the Iberian lynx in Spain. The presence of rodent/leporid-adapted *Cryptosporidium* species in the surveyed animals suggests spurious infections associated to the Iberian lynx’s diet. The Iberian lynx seems a suitable host for zoonotic genetic variants of *Cryptosporidium* (*C. parvum*) and *G. duodenalis* (assemblages A and B), although the potential risk of human transmission is regarded as limited due to light parasite burdens and suspected low excretion of infective (oo)cysts to the environment by infected animals. More research should be conducted to ascertain the true impact of these protozoan parasites in the health status of the endangered Iberian lynx.

## 1. Introduction

*Cryptosporidium* spp. and *Giardia duodenalis* are major causative agents of diarrheal diseases in humans and a wide diversity of animals with a worldwide distribution [1,2]. Human cryptosporidiosis is the leading protozoan cause of diarrheal mortality worldwide [3]. In contrast, human giardiasis is rarely mortal but is associated with malabsorptive diarrhoea and impaired childhood growth [4,5]. Both cryptosporidiosis and giardiasis also cause diarrhoea in neonatal ruminants, leading to high morbidity and mortality rates in the first three weeks [6,7,8,9] and significant economic losses for farmers [10,11]. *Cryptosporidium* and *Giardia* infections are typically asymptomatic in free-living animals, raising concerns about their true health impact in wildlife and the role of wildlife in the epidemiology of these parasites [6,12].

To date, at least 44 recognised *Cryptosporidium* species and more than 120 genotypes have been described. Of them, 19 species and four genotypes have been reported in humans with anthroponotic *C. hominis*, zoonotic *C. parvum*, avian-adapted *C. meleagridis*, canine-adapted *C. canis*, and feline-adapted *C. felis* being the most prevalent [13]. The epidemiology of *Cryptosporidium* infections in free-living carnivore species is poorly understood. In Europe, at least 11 *Cryptosporidium* species (*C. alticolis*, *C. andersoni*, *C. bovis*, *C. canis*, *C. ditrichi*, *C. erinacei*, *C. felis*, *C. hominis*, *C. parvum*, *C. suis*, and *C. ubiquitum*) and four genotypes (mink genotype, muskrat genotype, skunk genotype, and vole genotype) have been identified in 18 free-living carnivore species belonging to 12 genera and six families in the last 20 years (Table 1). The skunk genotype (24.2%, 32/132), *C. canis* (18.9%, 25/132), and *C. ditrichi* (16.7%, 22/132) were the most prevalent *Cryptosporidium* genetic variants found, whereas the red fox (*n* = 770) and the raccoon (*n* = 165) were the most investigated carnivore host species (Table 1) [14,15,16,17,18,19,20,21,22,23,24,25].

There are nine validated *Giardia* species in various vertebrates, namely *G. agilis* in amphibians; *G. ardeae* and *G. psittaci* in birds; *G. cricetidarum*, *G. microti*, *G. muris*, and *G. paramelis* in rodents; *G varani* in reptiles; and *G. duodenalis* in mammals including humans [13]. *Giardia duodenalis* is now regarded as a multispecies complex comprising eight established genotypes, known as assemblages A to H, that likely represent different species [27]. Five distinct *G. duodenalis* assemblages, zoonotic A and B, canine-adapted C and D, and ungulate-adapted E, have been identified in 20 European carnivore species belonging to 13 genera and seven families in the last two decades (Table 2) [27,28,29,30,31,32,33,34,35,36,37,38].

Assemblages B (34.8%, 23/66), A (30.3%, 20/66), and D (18.2%, 12/66) were the most prevalent *G. duodenalis* genetic variants individually found, whereas the red fox (*n* = 1129) and the wolf (*n* = 264) were the most investigated carnivore host species (Table 2).

The Iberian lynx (*Lynx pardinus*) is an emblematical felid species endemic to the Iberian Peninsula. It is listed as “endangered” by the International Union for Conservation of Nature’s Red List of Threatened Species [42]. Since the second half of the twentieth century, a sharp decrease in the number of Iberian lynxes brought the species to the brink of extinction due to habitat loss/transformation, illegal hunting, road kills, reduction in the density of its primary prey, the European rabbit (*Oryctolagus cuniculus*), and infectious diseases [43,44]. Among the latter, clinical cases and mortality reported during the last two decades have been associated to bacterial (e.g., *Mycobacterium bovis*, *Streptococcus canis*) [45,46], viral (e.g., feline leukaemia virus, feline herpes virus, feline calicivirus, pseudorabies virus) [47,48], and parasitic (e.g., *Neospora caninum*, *Toxoplasma gondii*, *Cystoisospora* spp.) [49,50,51,52] pathogens. Although the development of conservation programs has reversed the trend, allowing for an increase in the Iberian lynx population to over 1600 free-ranging animals in 2022 [53], the monitoring of pathogens that could affect captive and free-ranging animals is still a key component of ongoing conservation programs [54,55]. Following this line of action, this study aims to investigate the occurrence, genetic diversity, and zoonotic potential of the diarrhoea-causing enteric protozoan *Cryptosporidium* spp. and *G. duodenalis* in the Iberian lynx, a host species for which this information is currently lacking.

## 2. Materials and Methods

### 2.1. Study Area and Sampling

Faecal samples (*n* = 251) from Iberian lynxes were collected between 2017 and 2023. These included a total of 223 free-ranging animals from the three major population nuclei of this species in Spain (central, *n* = 63; south, *n* = 125; southwest, *n* = 33; unknown, 2), whereas 20 were lynxes maintained in captivity, including 14 animals from three captive breeding centres (BC1–BC3) belonging to the Iberian lynx ex situ conservation program and six from four zoo/conservation centres (ZC1–ZC4). The breeding and zoo/conservation centres were located in southern (*n* = 9) and southwestern (*n* = 10) Spain, respectively (Figure 1). Status information was not available for eight animals. In addition, five (three free-living, two captive) animals were longitudinally sampled during the study period. All faecal samples were taken from biological banks or animals subjected to medical check-ups, health programs, or surgical interventions during the study period. Faecal samples were obtained from the ground or the intestinal content of examined animals. Epidemiological information, including habitat status (free-living vs. captivity), sampling date, age (yearlings: <1 year old; subadults: 1 to 3 years old; adults: 3 to 10 years old; senile: >10 years old), sex, and sampling georeferenced location, was collected from each animal, whenever possible. All faecal samples studied were formed. This survey expands and complements those previously conducted on the very same Iberian lynx population that investigated the presence of other intestinal protists, including Microsporidia [56] and *Blastocystis* sp. (Caballero-Gómez et al., under preparation).

### 2.2. DNA Extraction and Purification of Faecal and Tissue Samples

Genomic DNA was isolated from approximately 100 mg of each faecal sample by using the IndiSpin Pathogen Kit (Indical Bioscience, Leipzig, Germany) according to the manufacturer’s instructions. Extracted and purified DNA samples were eluted in 90 µL of PCR-grade water and kept at 4 °C until further molecular analysis.

### 2.3. Molecular Detection and Characterisation of Cryptosporidium spp.

*Cryptosporidium* spp. presence was investigated using a nested PCR protocol, amplifying a 587 bp fragment of the small subunit of the rRNA (*ssu* RNA) gene of the parasite [57]. A subtyping tool based on the amplification of partial sequences of the 60 kDa glycoprotein (*gp60*) [58] gene was used to ascertain intra-species genetic diversity in the samples that tested positive for *C. parvum* and *C. cuniculus* with *ssu*-PCR.

### 2.4. Molecular Detection and Characterisation of Giardia duodenalis

For the identification of *G. duodenalis*, a real-time PCR (qPCR) method was set-up to amplify a 62 bp fragment of the *ssu* RNA gene of the parasite [59]. Samples that yielded cycle threshold (C_T_) values < 35 in qPCR were then analysed through a nested PCR, used to amplify a 300 bp fragment of the *ssu* RNA gene [60,61] to assess *G. duodenalis* molecular diversity at the assemblage level. Samples that yielded qPCR C_T_ values < 32 were additionally assessed using a sequence-based multilocus genotyping (MLST) scheme targeting the genes encoding for the glutamate dehydrogenase (*gdh*), β-giardin (*bg*), and triose phosphate isomerase (*tpi*) proteins to assess *G. duodenalis* molecular diversity at the sub-assemblage level. A 432 bp fragment of the *gdh* gene was amplified using a semi-nested PCR [62], while 511 and 530 bp fragments of the *bg* and *tpi* genes, respectively, were amplified through nested PCRs [63,64].

### 2.5. General Procedures

Detailed information on the PCR cycling conditions and oligonucleotides used for molecular identification and/or characterisation of the abovementioned parasites can be found in Appendix A, respectively. The previously described PCR protocols were conducted on a 2720 Thermal Cycler (Applied Biosystems, Foster City, CA, USA). The reaction mixes included 2.5 units of MyTAQ^TM^ DNA polymerase (Bioline GmbH, Luckenwalde, Germany) and 5–10 μL 5× MyTAQ^TM^ Reaction Buffer containing five mM deoxynucleotide triphosphates and 15 mM MgCl_2_. Negative and positive controls were included in all PCR runs. The PCR amplicons obtained were examined on a 1.5% D5 agarose gel stained with Pronasafe (Conda, Madrid, Spain) and sized using a 100 bp DNA ladder (Boehringer Mannheim GmbH, Mannheim, Germany).

### 2.6. Sequence and Phylogenetic Analysis

All amplicons of the expected size were directly sequenced in both directions with the internal primer pair in 10 μL reactions using Big Dye^TM^ chemistries and an ABI 3730xl sequencer analyser (Applied Biosystems). The raw sequencing data were examined with Chromas Lite version 2.1 software (http://chromaslite.software.informer.com/2.1, accessed on 18 January 2023) to generate consensus sequences. These sequences were compared with reference sequences deposited at the National Center for Biotechnology Information (NCBI) using the BLAST tool (http://blast.ncbi.nlm.nih.gov/Blast.cgi, accessed on 18 January 2023).

To analyse the phylogenetic relationship among *Cryptosporidium* species and genotypes at the *ssu* rRNA locus, a maximum-likelihood tree was constructed using MEGA version 10 [65], based on substitution rates calculated with the general time reversible model and gamma distribution with invariant sites (G+I). Bootstrapping with 1000 replicates was used to determine support for the clades. The representative nucleotide sequences obtained in the present study were deposited in the GenBank public repository database under accession numbers OR916202-OR916206 and OR921171 (*Cryptosporidium* spp.) and OR916207-OR916209 and OR921172-OR921177 (*G. duodenalis*).

### 2.7. Statistics Analysis

Prevalence rates were estimated by dividing the number of positive animals by the total number of animals tested using two-sided exact binomial 95% confidence intervals (95% CI). Pearson’s chi-squared test or Fisher’s exact test was used to assess differences in the *Cryptosporidium* spp. and *G. duodenalis* infection rates, according to habitat, sex, age, sampling areas, and sampling period (categorised by terciles), using the R Statistical Package version 2.15.3 [66]. A *p*-value < 0.05 was considered as statistically significant.

## 3. Results

The full dataset of this study, showing sampling, epidemiological, diagnostic, and molecular data, can be found in Appendix A.

### 3.1. Occurrence of Cryptosporidium spp. and Giardia duodenalis

Table 3 summarises the occurrence of *Cryptosporidium* spp. and *G. duodenalis* in the Iberian lynx population (*n* = 251) under investigation according to the main epidemiological variables considered in this study. All faecal samples analysed (*n* = 256) had a formed consistency, suggestive of an apparent absence of gastrointestinal manifestations.

*Cryptosporidium* spp. DNA was detected in 2.4% (6/251; 95% CI: 0.9–5.1) of the individuals tested. On the other hand, *G. duodenalis* DNA was detected in 27.9% (70/251; 95% CI: 22.4–33.9) of the individuals tested. *Giardia* infections were observed in animals of all age groups, whereas no *Cryptosporidium* infections were detected in senile individuals. Three Iberian lynxes (two free-living, one captive) were co-infected with *Cryptosporidium* spp. and *G. duodenalis*.

None of the epidemiological variables considered were significantly associated with a higher likelihood of *Giardia* or *Cryptosporidium* infection except the sampling period for the latter *(p* = 0.042). The highest prevalence was detected in individuals sampled during the 2017–2020 period (6.8%), followed by 2022–2023 (1.8%) and 2021 (0.0%). Both *Cryptosporidium* spp. and *G. duodenalis* were detected in the three free-ranging areas sampled with frequencies varying from 1.5 to 7.3% and 23.9 to 33.3%, respectively.

### 3.2. Molecular Characterisation of Cryptosporidium spp.

The sequence analyses of the *ssu* rRNA region revealed the presence of four distinct *Cryptosporidium* species (*C. alticolis*, *C. cuniculus*, *C. occultus*, and *C. parvum*) in the Iberian lynx populations under study (Table 4). *Cryptosporidium alticolis* was identified in a free-living animal from south Spain. The sequences generated at the *ssu* rRNA locus differed by five single nucleotide polymorphisms (SNPs, including three indels) from reference sequence MH145330 originally isolated from a common vole in the Czech Republic. *Cryptosporidium cuniculus* was identified in a free-living and a captive Iberian lynx, both in south Spain (Table 4 and Figure 1). *Ssu* rRNA sequences had 100% identity with reference sequence AY120901. One of the two isolates was successfully genotyped at the *gp60* locus, revealing the presence of genotype VaA19. *Cryptosporidium occultus* was identified in a free-living animal in southwest Spain. Two additional isolates were assigned to *C. parvum* at the *ssu* rRNA marker: one belonged to a captive Iberian lynx in central Spain and the other to a free-living animal in southwest Spain. Both *ssu* rRNA sequences differed by 6–7 SNPs from the reference sequence AF112571 (Table 4). These include a hallmark deletion of 3–4 nucleotides at positions 686 to 689 of AF112571. Attempts to amplify these sequences at the *gp60* marker failed.

Phylogenetic analysis of *ssu* rRNA sequences revealed that all sequences generated in the present study belonging to *C. alticolis*, *C. cuniculus*, and *C. parvum* grouped together with appropriate reference sequences in well-defined clusters (Figure 2).

### 3.3. Molecular Characterisation of Giardia duodenalis

*Giardia*-positive samples with qPCR yielded C_T_ values ranging from 20.0 to 39.7 (median: 34.5; standard deviation: 3.5). Approximately half of them (53.0%, 35/66) had C_T_ values > 34 and were not further investigated for genotyping purposes. All 31 *Giardia*-positive samples with qPCR C_T_ values ≤ 34 were subjected to nested *ssu*-PCR to ascertain the assemblage of the parasite involved. Of them, 25.8% (8/31) were successfully genotyped at this locus (Table 5). Sequence analyses revealed that assemblage A (62.5%, 5/8) was more prevalent than assemblage B (37.5%, 3/8). Overall, MLST data at the four assessed loci were available for 3.0% (2/66) of samples, whereas subtyping data at a single locus (*ssu* rRNA) were available for 9.1% (6/66) of samples. No mixed infections nor host-adapted assemblages of canine (C, D), feline (F), or livestock (E) origin were detected.

Out of the three assemblage A sequences at the *ssu* rRNA locus, two showed 100% identity with reference sequence M54878 with the remaining one differing from it by three SNPs in the form of ambiguous (double peak) positions. A single assemblage A sequence was confirmed as sub-assemblage AI at the *gdh*, *bg*, and *tpi* loci. The sequences generated at the three markers were identical to their respective reference sequences (Table 6).

All three assemblage B sequences at the *ssu* rRNA locus showed 100% identity with reference AF113898. One of them was successfully genotyped at the three markers used, being identified as sub-assemblage BIV at the gdh marker and as sub-assemblage BIII at the *tpi* marker. This sample was, therefore, considered as an ambiguous BIII/BIV isolate (Table 6).

## 4. Discussion

This study shows that *Cryptosporidium* spp. and *G. duodenalis* are present at very different rates (2.4% vs. 27.9%) in faecal samples from Iberian lynxes without apparent gastrointestinal manifestations. The strengths of this study include (i) the use of molecular (PCR and Sanger sequencing) methods for accurate detection and genotyping of the two pathogens under investigation, (ii) a large sample size that includes a significant proportion (15–20%) of the estimated total population of free-living Iberian lynxes, (iii) representativeness of all three major distribution areas where the Iberian lynx is naturally present in Spain, (iv) the first report describing the molecular diversity of *Cryptosporidium* spp. and *G. duodenalis* in this carnivore host species, and (v) molecular evidence suggesting that a significant proportion of the positive samples might correspond to spurious infections as a direct consequence of predation on infected preys.

Cryptosporidiosis is regarded as a high-risk and often fatal opportunistic infection for undernourished young children and immunocompromised individuals as well as a major cause of neonatal diarrhoea in livestock [1,2,3]. Comparatively, much less information is available on the epidemiology of *Cryptosporidium* spp. in wildlife with most studies conducted globally indicating low-to-medium infection rates and an apparent absence of gastrointestinal manifestations [2]. This trend is particularly manifest in wild carnivore species. In the European scenario, *Cryptosporidium* infections have been reported in badgers (2.8–20.0%), foxes (6.1–13.3%), genets (16.6%), Eurasian lynxes (4.2%), martens (29.2–29.4%), minks (6.2%), otters (4.0%), raccoons (3.9–43.7%), raccoon dogs (24.1%), and wolves (35.7%), mostly with PCR (Table 1). Only two previous studies conducted in the Iberian Peninsula attempted to identify the presence of *Cryptosporidium* spp. in Iberian lynxes, but the limited number of samples analysed did not allow for the detection of the protozoa [20,32]. In the present survey, *Cryptosporidium* spp. was detected in 2.4% (6/251) of the faecal samples from the Iberian lynxes examined, a figure in the lower range of those reported for other free-living carnivore species in Spain, Portugal, and other European countries. Despite the limited prevalence, positive animals were detected in the three sampling areas. These findings, together with the statistically significant differences among sampling periods, denote a wide but temporally heterogeneous circulation of *Cryptosporidium* in the Iberian lynx populations.

Molecular analyses of the six *Cryptosporidium*-positive isolates successfully genotyped revealed interesting data. First, four of the six infections detected were caused by *Cryptosporidium* species (*C. alticolis*, *C. cuniculus*, and *C. occultus*) with a strong preference for hosts that are common preys of the Iberian lynx. In this regard, although the Iberian lynx diet is mainly based on European rabbit, they can sporadically consume birds, wild ungulates, and also small mammals [67]. Rodent-adapted *Cryptosporidium alticolis* and *C. occultus* were initially described in common voles and rats [68,69], whereas leporids, including rabbits and hares, are the preferred host species for *C. cuniculus* [70]. Interestingly, *C. alticolis* has been previously reported in two red foxes in Poland [17]. To our knowledge, this is the first report of *C. cuniculus* and *C. occultus* in free-living carnivores (including the Iberian lynx) globally. Taken together, these data seem to indicate that the presence of *C. alticolis*, *C. cuniculus*, and *C. occultus* in faecal samples from Iberian lynxes might be the consequence of spurious (mechanical carriage) rather than true infections. Second, the identification of generalist *C. parvum* allows for a wider interpretation. This *Cryptosporidium* species is characterised by a loose host specificity and great cross-species potential [71], making difficult the distinction between spurious and true infections. Regardless the case, the failure to amplify the two *C. parvum* isolates at the *gp60* marker might be indicative of a low number of oocysts in faeces, compatible with a subclinical infection. *Cryptosporidium parvum* infections have been described in other European free-living carnivores, including wolves in Poland [15] and red foxes in Spain [20,21] and the UK [22]. Third, we managed to characterise one of our two *C. cuniculus* isolates as genotype VaA19. Of note, previous studies conducted in Spain reported the presence of VaA16 (*n* = 1), VaA18 (*n* = 2), VbA24 (*n* = 1), VbA26 (*n* = 1), and VbA31 (*n* = 1) in wild populations of European rabbits and Iberian hares [72,73]. These data expand our knowledge on the epidemiology of *C. cuniculus* in the country and support the spurious nature of our findings in Iberian lynxes. And fourth, the assignment of one of our *Cryptosporidium*-positive isolates as *C. occultus* should be interpreted with caution, as the generated *ssu* sequence was relatively short (214 bp) and this species is closely related to *C. suis* [69]. We based our decision on two facts: (i) Our *C. occultus* sequence differed by two SNPs (688DelA, and T692A) with *C. suis* reference sequence AF115377, and (ii) the predator–prey relationship makes more likely that Iberian lynxes fed on small rodents than on suids, including domestic pigs and wild boars (the natural host species for *C. suis*).

Our molecular findings on the frequency and diversity of *Cryptosporidium* species in the Iberian lynx could also have public health implications. Whereas *C. alticolis* is not considered a zoonotic pathogen and only sporadic cases of human cryptosporidiosis by *C. occultus* have been reported in China [74], both *C. cuniculus* and *C. parvum* are able to cause significant morbidity in humans. *Cryptosporidium cuniculus* is typically identified at low infection (<1.5%) rates in European countries, including Spain [75], Sweden [76], and the UK [77,78]. However, because *C. cuniculus* is closely related to *C. hominis*, its potential to cause human infections if the opportunity arises should not be underestimated [79]. The finding of *C. parvum* has more relevance as this *Cryptosporidium* species causes one in four human cryptosporidiosis cases in Spain [80,81,82,83,84,85].

In contrast with cryptosporidiosis, giardiasis is widely regarded as a debilitating rather than a fatal condition in both human [5,86] and animal [9,10] hosts. Giardiasis in free-ranging animals has only been investigated opportunistically, and relatively little is known about the epidemiology and health impact of the infection in wildlife populations [87]. At the European level, *Giardia* infections have been reported in several wild carnivores, including badgers (25.6%), jackals (12.5%), lynxes (16.7%), martens (12.5–15.8%), otters (3.1–6.8%), raccoons (29.2–33.3%), red foxes (2.2–44.2%), wildcats (10.0%), and wolves (5.0–28.6%), mostly with PCR (Table 2). An infection rate of 26.7% was reported in 30 Iberian lynxes sampled from Portugal in a previous study [32], a figure very similar to that (27.9%) found in the present study also with PCR. These data suggest a high circulation of this parasite among the Iberian lynx populations and denote that the Iberian lynx could be a suitable host for *G. duodenalis*. The fact that neither geographical origin, sex, age, status, nor sampling year have an effect on the likelihood of having the parasite seems to support this hypothesis.

In the present study, the effort to assess the genetic diversity of *G. duodenalis* was hampered by the limited amount of parasitic DNA present in most positive samples, as indicated by the median qPCR C_T_ value (34.5). This fact compromised the performance of our genotyping PCRs and explains why only a low proportion (25.8%, 8/31) of the tested *G. duodenalis*-positive samples were successfully characterised at one or more of the four (*ssu*, *gdh*, *bg*, and *tpi*) genetic markers used for this purpose. Our sequence analyses revealed the presence of two assemblages with assemblage A being more prevalent than assemblage B (62.5% vs. 37.5%, respectively). Remarkably, no feline-specific assemblage F was identified in the surveyed Iberian lynx populations. Considering that both assemblages A and B have zoonotic potential, these findings deserve attention. Out of the six assemblage A sequences, only one could be resolved at the sub-assemblage level as AI. This sub-assemblage is the most frequently found in animals [13], although it has also been reported at non-negligible rates in some human communities, primarily in low-income countries [88]. The finding of assemblage B is somehow more worrying as this genetic variant is the most predominantly found circulating in the Spanish human population regardless of clinical status [89,90,91,92]. Of note, in the only survey reporting molecular data on *G. duodenalis* infections in European free-living felids, assemblage B was identified in a single wildcat in Luxembourg [36]. Taken together, these findings indicate that felids including the Iberian lynx can act as suitable hosts and spreaders of zoonotic variants of *G. duodenalis*. However, the finding that *G. duodenalis* infections are most likely associated with light parasite burdens (and, therefore, low cyst count in faeces) might limit the environmental contamination with infective cysts and reduce human exposure to them.

This study has some limitations that should be considered when interpreting the results obtained. First, it is possible that long-term storage of faecal samples has affected the quality/quantity of parasitic DNA, reducing the sensitivity and compromising the performance of the PCR protocols used for detection and genotyping purposes. Second, light parasitic infections leading to low (oo)cyst counts in faecal samples together with the limited sensibility of our genotyping PCRs have negatively impacted our ability to determine intra-species molecular variability in some *Cryptosporidium*- and *G. duodenalis*-positive samples. And third, low *Cryptosporidium* infection rates might have compromised the accuracy of the statistical analyses conducted.

## 5. Conclusions

This study describes for the first time the occurrence and genetic diversity on *Cryptosporidium* spp. and *G. duodenalis* in the endangered Iberian lynx. The large sample size available, including animals from the main distribution areas, guarantee that the results obtained are representative of the whole free-living Iberian lynx population in Spain. Our results denote a limited but wide circulation of *Cryptosporidium* and a high wide and endemic distribution of *Giardia* among these individuals, which could be of animal health concern. The generated molecular data suggest that most *Cryptosporidium* species found correspond to rodent- or leporid-adapted strains that very likely cause spurious rather than true infection in the surveyed Iberian lynxes. However, the finding of zoonotic *C. parvum* and *G. duodenalis* assemblages A and B indicates that the Iberian lynx can act as a suitable host and spreader of these pathogens. Although the role of the Iberian lynx as a source of human cryptosporidiosis and giardiasis is regarded as low, this possibility should not be underestimated. Individuals (researchers, veterinarians, hunters) in close contact with infected animals or their faeces should be aware of the potential risk of zoonotic transmission of these protozoan parasites. The information provided in this study expands our knowledge on the epidemiology and public health relevance of *Cryptosporidium* spp. and *G. duodenalis* in Spain.

## Figures and Tables

**Figure 1 animals-14-00340-f001:**
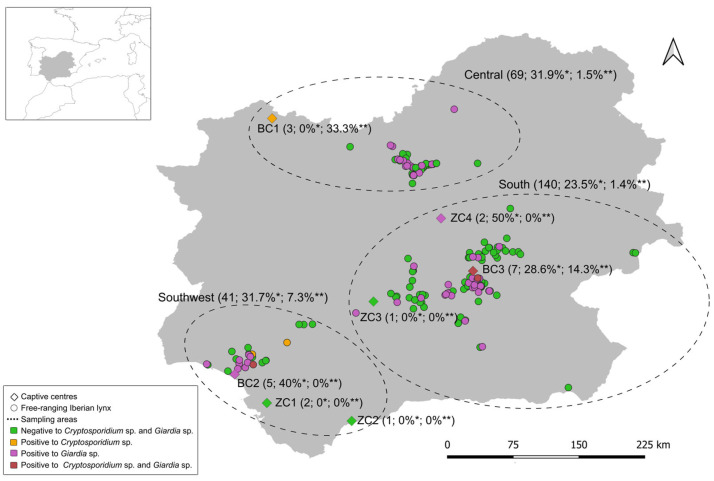
Spatial distribution and molecular results of Iberian lynx samples. Total number of faecal samples analysed (*n* = 256) and frequency of positivity of *Giardia duodenalis* (*) and *Cryptosporidium* spp. (**) in each sampling area and captivity centre are shown in brackets.

**Figure 2 animals-14-00340-f002:**
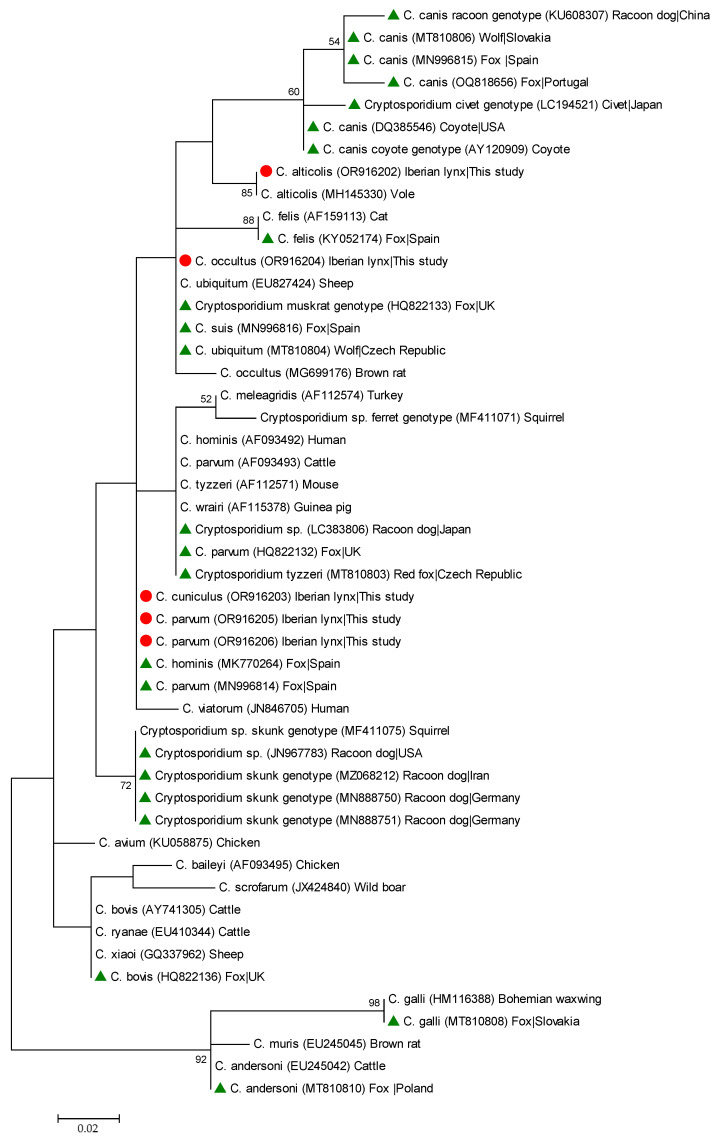
Phylogenetic relationship among *Cryptosporidium* species and genotypes revealed with a maximum likelihood analysis of the partial *ssu* rDNA gene. Substitution rates were calculated by using the general time reversible model. Numbers on branches are percent bootstrapping values over 50% using 1000 replicates. The filled red circle indicates the nucleotide sequence generated in the present study. The filled green triangle indicates selected nucleotide sequences previously reported in wild carnivore species globally used for comparative purposes.

**Table 1 animals-14-00340-t001:** Infection rates and molecular diversity of *Cryptosporidium* spp. in European wild carnivore species, 2007–2023.

Family	Host (Common Name)	Host (Scientific Name)	Country	Frequency (%)	No. pos./Total	Genotype(s) (*n*)	Reference
Canidae	Arctic fox	*Vulpes lagopus*	Norway	0.0	0/62	*–*	[14]
	Grey wolf	*Canis lupus*	Poland	35.7	5/14	*C. parvum* genotype 2 (5)	[15]
	Iberian wolf	*Canis lupus signatus*	Portugal	2.5	3/121	*C. canis* (3)	[16]
	Raccoon dog	*Nyctereutes procyonoides*	Poland	24.1	21/87	*C. canis* (dog genotype) (16)	[17]
	Red fox	*Vulpes vulpes*	Ireland	0.0	0/13	*–*	[18]
			Norway	0.0	0/269	*–*	[19]
			Poland	12.0	6/50	*C. canis* (fox genotype) (3), *C. alticolis* (2), *C. vole* genotype II (1)	[17]
			Portugal	3.3	4/121	*C. canis* (4)	[16]
			Spain	8.0	7/87	*C. canis* (2), *C. felis* (1), *C. parvum* (3), *C. ubiquitum* (1)	[20]
				6.1%	12/197	*C. hominis* (4), *C. canis* (3), *C. parvum* (2), *C. ubiquitum* (1), *C. suis* (1)	[21]
			UK	8.0	10/124	*C. parvum* (2)	[22]
			UK	13.3	4/30	*C. bovis* (1), *C. parvum* (1), *C. muskrat* genotype II (1)	[23]
Felidae	Eurasian lynx	*Lynx lynx*	Germany	4.2	1/24	*C. felis* (1)	[24]
	Iberian lynx	*Lynx pardinus*	Portugal	3.3	1/30	*C. felis* (1)	[16]
			Spain	0.0	0/6	*–*	[20]
	Wildcat	*Felis silvestris*	Spain	0.0	0/2	*–*	[20]
Herpestidae	Mongoose	*Herpestes ichneumon*	Spain	50.0	1/2	*C. canis* (1)	[20]
Mustelidae	American mink	*Mustela vison*	Ireland	6.2	5/81	*C. mink* genotype (1), *C. andersoni* (3), *Cryptosporidium* spp. (1)	[18]
	Beech marten	*Martes foina*	Poland	29.4	15/51	*C. ditrichi* (15)	[17]
			Spain	0.0	0/8	*–*	[20]
	Eurasian badger	*Meles meles*	Ireland	0.0	0/7	*–*	[18]
			Poland	20.0	9/45	*C. skunk* genotype (5), *C. erinacei* (4)	[17]
			Spain	2.8	2/70	*C. hominis* (1), *Cryptosporidium* spp. (1)	[20]
	Eurasian otter	*Lutra lutra*	Ireland	4.0	1/25	*Cryptosporidium* spp. (1)	[18]
			Spain	0.0	0/2	*–*	[20]
	Ferret	*Mustela patois furo*	Spain	0.0	0/2	*–*	[20]
	Genet	*Genetta genetta*	Spain	16.6	1/6	*Cryptosporidium* spp. (1)	[20]
	Irish stoats	*Mustela ermine* *hibernica*	Ireland	0.0	0/30	*–*	[18]
	Pine marten	*Martes martes*	Poland	29.2	7/24	*C. ditrichi* (7)	[17]
			Ireland	0.0	0/7	*–*	[18]
	Polecat	*Mustela putorius*	Spain	0.0	0/2	*–*	[20]
Procyonidae	Raccoon	*Procyon lotor*	Poland	24.6	16/65	*C. skunk* genotype (16)	[17]
			Poland	43.7	14/32	*C. skunk* genotype (9), *Cryptosporidium* spp. (5)	[25]
			Germany	3.9	2/51	*C. skunk* genotype (2)	[26]
			Germany	17.6	3/17	*Cryptosporidium* spp. (3), *C. erinacei* (3), *C. suis* (2)	[25]

**Table 2 animals-14-00340-t002:** Infection rates and molecular diversity of *Giardia duodenalis* in European wild carnivore species, 2007–2023.

Family	Host (Common Name)	Host (Scientific Name)	Country	Frequency (%)	No. pos./Total	Genotype(s) (*n*)	Reference
Canidae	Apennine wolf	*Canis lupus italicus*	Italy	5.0	1/20	C (1)	[28]
				100	1/1	D (1)	[29]
	Grey wolf	*Canis lupus*	Croatia	10.2	13/127	A (1), A1 (5), C (2), D (1), AI+B+D (1), A+C+D (1), C+D (1)	[30]
			Poland	28.6	2/7	D (2)	[31]
			Romania	100	3/3	D (3)	[32]
	Iberian wolf	*Canis lupus signatus*	Portugal	25.6	31/121	D (4), C+D (2)	[16]
			Spain	15.9	1/6	Unknown	[20]
	Jackal	*Canis aureus*	Croatia	12.5	1/8	A+B (1)	[30]
	Raccoon dog	*Nyctereutes* *procyonoides*	Romania	100	1/1	D (1)	[32]
	Red fox	*Vulpes vulpes*	Croatia	4.6	3/66	A (1)	[30]
			Italy	7.0	5/71	Unknown	[33]
			Norway	2.2	6/269	A (3), AI (2), B3 (1)	[19]
			Portugal	18.6	22/118	C+D (1)	[16]
			Romania	4.6	10/217	A (2), B (1)	[34]
			Spain	8.1	7/87	Unknown	[20]
				9.6	19/197	Unknown	[21]
			Sweden	44.2	46/104	B (4)	[35]
Felidae	Eurasian lynx	*Lynx lynx*	Germany	16.7	4/24	Unknown	[24]
	Iberian lynx	*Lynx pardinus*	Portugal	26.7	8/30	Unknown	[16]
			Spain	0.0	0/6	–	[20]
	Wildcat	*Felis silvestris*	Luxembourg	10.0	1/10	B (1)	[36]
				0.0	0/2	–	[20]
Herpestidae	Mangoose	*Herpestes ichneumon*	Spain	0.0	0/2	–	[20]
Mustelidae	Badger	*Meles meles*	Italy	25.6	11/43	AII (6)	[39]
			Poland	0.0	0/1	–	[31]
			Spain	0.0	0/70	–	[20]
			UK	100	1/1	E (1)	[40]
	Ferret	*Mustela putorius furo*	Spain	0.0	0/2	–	[20]
	Marten	*Martes* sp.	Poland	0.0	0/1	–	[31]
	Eurasian otter	*Lutra lutra*	Denmark	3.1	1/33	Unknown	[37]
			Poland	0.0	0/1	–	[31]
			Spain	6.8	30/437	Unknown	[38]
				0.0	0/2	–	[20]
	Polecat	*Mustela putorius*	Spain	0.0	0/2	–	[20]
	Stone marten	*Martes foina*	Portugal	15.8	3/19	Unknown	[32]
			Spain	12.5	1/8	Unknown	[20]
	Weasel	*Mustela* sp.	Poland	0.0	0/1	–	[31]
Procyonidae	Racoon	*Procyon lotor*	Luxembourg	33.3	3/9	B (3)	[41]
			Germany	29.2	14/48	B (13)	[41]
Ursidae	Brown bear	*Ursus arctos*	Croatia	0.0	0/19	–	[30]
Viverridae	Genet	*Genetta genetta*	Spain	0.0	0/6	–	[20]

**Table 3 animals-14-00340-t003:** Infection rates by *Cryptosporidium* spp. and *Giardia duodenalis* in Iberian lynxes (*n* = 251) according to distribution area, sex, age, status, and sampling year of the animals. 95% confidence intervals (95% CI) are indicated.

		*Cryptosporidium* spp. (*n* = 6)			*Giardia duodenalis* (*n* = 70)		
Variable	Animals (*n*)	Positive (*n*)	% (95% CI)	*p*-Value	Positive (*n*)	% (95% CI)	*p*-Value
Sampling area (6) ^a^							
Central	66	1	1.5 (0.04–8.2)	0.101	22	33.3 (22.2–46.0)	0.307
South	138	2	1.5 (0.2–5.1)		33	23.9 (17.1–31.9)	
Southwest	41	3	7.3 (1.5–19.9)		13	31.7 (18.1–48.1)	
Sex (87) ^a^							
Male	95	2	2.1 (0.3–7.4)	0.619	21	22.1 (14.2–31.8)	0.424
Female	69	1	1.5 (0.04–7.8)		19	27.5 (17.5–39.6)	
Age (67) ^a.b^							
Yearling	54	2	3.7 (0.5–12.8)	0.624	13	24.1 (13.5–37.6)	0.856
Sub-adult	77	1	1.3 (0.03–7.0)		21	27.3 (17.7–38.6)	
Adult	42	2	4.8 (0.6–16.2)		12	28.6 (15.7–44.6)	
Senile	11	0	0.0 (0.0–0.0)		4	36.4 (10.9–69.2)	
Status (8) ^a^							
Free-living	223	4	1.8 (0.5–4.5)	0.079	64	28.7 (22.9–35.1)	0.476
Captive	20	2	10.0 (1.2–31.7)		5	25.0 (8.7–49.1)	
Sampling year (14) ^a^							
2017–2020	59	4	6.8 (1.9–16.5)	0.042	17	28.8 (17.8–42.1)	0.777
2021	69	0	0.0 (0.0–0.0)		17	24.6 (15.1–36.5)	
2022–2023	109	2	1.8 (0.2–6.5)		32	29.4 (21.0–38.9)	

^a^ Missing values (number of samples with unknown data). ^b^ yearlings: <1 year old; sub-adults: 1 to 3 years old; adults: 3 to 10 years old; senile: >10 years old.

**Table 4 animals-14-00340-t004:** Diversity, frequency, and molecular features of *Cryptosporidium* spp. isolates identified in the Iberian lynx population investigated in the present study.

Species	Genotype	Isolates (*n*)	Locus	Reference Sequence	Stretch	Single Nucleotide Polymorphisms	GenBank ID
*C. alticolis*	–	1	*ssu* rRNA	MH145330	311–781	A411T, 425_426DelTA, Ins464_467TAAT, 569DelT, 782InsG	OR916202
*C. cuniculus*	–	2	*ssu* rRNA	AY120901	319–784	None	OR916203
	VaA19	1	*gp60*	KU852733	5–750	None	OR921171
*C. occultus*	–	1	*ssu* rRNA	MG699176	482–695	None	OR916204
*C. parvum*	–	1	*ssu* rRNA	AF112571	528–1025	A646G, T649G, 686_689DelTAAT, A691T, A854R, A892G	OR916205
	–	1	*ssu* rRNA	AF112571	528–1030	646G, T649G, 686_688DelTAA, A691T, C795T, A891G, A933G	OR916206

Del: base deletion; *gp60*: 60 kDa glycoprotein; R: A/G; *ssu* rRNA: small subunit ribosomal RNA; Y: C/T.

**Table 5 animals-14-00340-t005:** Multilocus sequence typing results of the eight *G. duodenalis*-positive samples successfully genotyped at any of the four loci investigated in the present survey. The age and gender of the infected Iberian lynxes are also shown.

**Sample ID**	Age (yrs.)	Sex	C_T_ Value in qPCR	*ssu* rRNA	*gdh*	*bg*	*tpi*	Assigned Genotype
1091	Sub-adult	Female	33.1	B	–	–	–	B
962	Unknown	Unknown	20.0	B	BIV	B	BIII	BIII/BIV
1034	Sub-adult	Female	32.7	A	–	–	–	A
1079	Yearling	Unknown	24.2	B	–	–	–	B
486D	Adult	Female	24.2	A	–	–	–	A
1004	Sub-adult	Unknown	30.1	A	–	–	–	A
948	Unknown	Unknown	24.7	A	–	–	–	A
83H	Sub-adult	Female	20.3	A	AI	AI	AI	AI

*bg*: β-giardin; *gdh*: glutamate dehydrogenase; *ssu* rRNA: small subunit ribosomal RNA; *tpi*: triose phosphate isomerase.

**Table 6 animals-14-00340-t006:** Diversity, frequency, and molecular features of *G. duodenalis* isolates identified in the Iberian lynx population investigated in the present study.

**Assemblage**	Sub-Assemblage	Isolates (*n*)	Locus	Reference Sequence	Stretch	Single Nucleotide Polymorphisms	GenBank ID
A	–	4	*ssu* rRNA	M54878	1–289	None	OR916207
	–	1	*ssu* rRNA	M54878	1–289	A87W, G153R, C207Y	OR916208
	AI	1	*gdh*	L40509	73–491	None	OR921172
	AI	1	*bg*	AY655702	27–521	None	OR921173
	AI	1	*tpi*	L02120	559–1072	None	OR921174
B	–	3	*ssu* rRNA	AF113898	1–275	None	OR916209
	BIV		*gdh*	L40508	89–490	T183C, C252T	OR921175
	–		*bg*	AY072727	98–593	None	OR921176
	BIII		*tpi*	AF069560	1–479	T134C, A176G, A395G	OR921177

*bg*: β-giardin; *gdh*: glutamate dehydrogenase; *ssu* rRNA: small subunit ribosomal RNA; *tpi*: triose phosphate isomerase.

## Data Availability

The authors confirm that the data supporting the findings of this study are available within the article and its Appendix A.

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
