# Peer review of "Detection and Molecular Diversity of Cryptosporidium spp. and Giardia duodenalis in the Endangered Iberian Lynx (Lynx pardinus), Spain"

_animals, 2024, doi:10.3390/ani14020340_

Round 1

Reviewer 1 Report

Comments and Suggestions for Authors

In this study (ID animals-2815750), the authors described the prevalence and diversity of Cryptosporidium and Giardia spp. in faeces coming from Lynxes between 2017 and 2023 in Spain. This research can contribute to understand the distribution of these parasitic agents and it has relevance since both can be zoonotic.

The manuscript is well-written and comprehensive. The results representation could be improved and the statistical analysis considering all variables. Additionally, I suggest representing the detection rates of each pathogen per year. The discussion touches on the main findings and the interpretation is adequate and interesting. In conclusion, it could be considered publishable after modifications.

Line 74, 75, …: Reference citation should be revised and modified. There are double citation (authors and numbers)

Table 1 and 2: I suggest reducing and focus both Tables to Lynx as host for Cryptosporidium and Giardia (objective of the manuscript).

Line 150: I would describe the number of samples collected/analysed considering the year, status of Lynxes. I suggest creating a table for describing the characteristic of samples collected.

Line 150: Please, indicate how faeces were collected (directly with a swab, from the ground, …)

Table 3: Have you considered statistically the impact of two variables together in the detection? For example, sex and year?

Please, what do you mean with missing values? Please, explain it in the manuscript.

Line 459: I suggest removing or to reformulate the sentence “the large sample size…. are representative of the whole-free living Iberian lynx population in Spain” It is too ambitious; the samples were collected in a period of 6 years and also captive status is included.

Author Response

In this study (ID animals-2815750), the authors described the prevalence and diversity of Cryptosporidium and Giardia spp. in faeces coming from Lynxes between 2017 and 2023 in Spain. This research can contribute to understand the distribution of these parasitic agents and it has relevance since both can be zoonotic. The manuscript is well-written and comprehensive. The results representation could be improved and the statistical analysis considering all variables. Additionally, I suggest representing the detection rates of each pathogen per year. The discussion touches on the main findings and the interpretation is adequate and interesting. In conclusion, it could be considered publishable after modifications.

Reply: We thank Reviewer #1 for his/her preliminary positive appraisal on our manuscript. Regarding your comment about presenting results by sampling year, we did so in our first analyses. However, because the number of animals sampled yearly varied greatly, we lose statistical power. In order to overcome this limitation, we re-analyse our data in sampling periods for which the number of sampled animals were more similar. In this way we reached figures that were statistically meaningful.

  1. Line 74, 75, …: Reference citation should be revised and modified. There are double citation (authors and numbers)

Reply: References mentioned in the main body of the text have been now presented numerically in agreement with the journal´s editing requirements.

  1. Table 1 and 2: I suggest reducing and focus both Tables to Lynx as host for Cryptosporidium and Giardia (objective of the manuscript).

Reply: We disagree with Reviewer´s #1 suggestion. Please note that molecular epidemiological information on the occurrence and genetic diversity of Cryptosporidium and Giardia in wild felid species have been previously investigated in only four studies (Mateo et al., 2017; Solarczyk et al. 2019; Segeritz et al. 2021; and Figueiredo et al. 2023). Current Tables 1 and 2 allows the direct comparison of data presented here with those obtained in other European mesocarnivore species, providing the right epidemiological frame to interpret the results obtained in the present study. A clear example of this is the conclusion that many of the Cryptosporidium and Giardia infections detected in the Iberian lynx population under study correspond to spurious rather than true infections, as it is also the case for other carnivore species in the European scenario.

  1. Line 150: I would describe the number of samples collected/analysed considering the year, status of Lynxes. I suggest creating a table for describing the characteristic of samples collected.

Reply: Please note that this information can be easily reached by the interested reader in current Table S3. The statistical significance of the variables mentioned (sampling area, sex, age group, status, and sampling year) have been thoroughly analysed in current Table 3. Additionally, the geographical distribution of positive and negative faecal samples can be seen in Figure 1.

  1. Line 150: Please, indicate how faeces were collected (directly with a swab, from the ground, …)

Reply: Following Reviewer´s 1 suggestion, the following sentence has been added in current lines 157-158: “Faecal samples were obtained from the ground or the intestinal content of examined animals”.

  1. Table 3: Have you considered statistically the impact of two variables together in the detection? For example, sex and year?

Reply: Following the reviewer suggestion, an interaction analysis among the variables included in the study was carried out. However, none of combinations were statistically associated with a higher risk of infection by Cryptosporidium or Giardia.

  1. Please, what do you mean with missing values? Please, explain it in the manuscript.

Reply: missing values are the data values that were not stored for a variable in the observation of interest (unknown values). This issue has been now clarified in the footnote of current Table 3.

  1. Line 459: I suggest removing or to reformulate the sentence “the large sample size…. are representative of the whole-free living Iberian lynx population in Spain” It is too ambitious; the samples were collected in a period of 6 years and also captive status is included.

Reply: We disagree with Reviewer´s #1 comment. Indeed, we believe that this is one of the main contributions of the present study. We should consider that we have analysed an estimate of 15–20% of the total population of living Iberian lynxes during the period of study. This is a huge achievement considering the difficulty of obtaining biological samples from an endangered species that was at the verge of extinction at the beginning of the 21st century, when less than 100 individuals survived in two isolated subpopulations in Andalusia (south of Spain) in 2002. Please also note that there are very few studies globally targeting such a significant proportion of total living animals within a given species. In our opinion this is a remarkable fact that must be adequately highlighted in the manuscript.

Reviewer 2 Report

Comments and Suggestions for Authors

This is an excellent MS! My remarks/suggestions are very few: A) Introduction. Additional explicit information on the Iberian lynx diet would be welcome; B) Material and Methods. Authors should describe in more detail the age classes that appear in Table 3; C) Discussion. I suggest that Authors discuss to which extent, in their opinion, the limited number of sampled cubs may have influenced the analysis of risk factors for Cryptosporidium infection. A comparison with similar surveys in other carnivore hosts would be welcome. A limited number of minor inaccuracies is highlighted in the attached file   

Author Response

This is an excellent MS! My remarks/suggestions are very few:

Reply: We thank Reviewer #2 for his/her preliminary positive appraisal on our manuscript.

  1. Additional explicit information on the Iberian lynx diet would be welcome;

Reply: The Iberian lynx diet is mainly based on European rabbit (Oryctolagus cuniculus), as indicated in current line 111. However, they can also sporadically consume birds (13.1%), and small mammals (6.7%) and wild ungulates (3.6%) (Gil-Sánchez et al. 2006). Additional information and this reference has been included in the Discussion section in lines 347-349. Consequently, this reference has been added to the Reference section.

  1. Material and Methods. Authors should describe in more detail the age classes that appear in Table 3

Reply: Age classes were divided as follows: yearlings: < 1 year old; subadults: 1 to 3 years old; adults: 3 to 10 years old; senile: > 10 years old. This information has been now added as a footnote of Table 3 in current line 239.

  1. I suggest that Authors discuss to which extent, in their opinion, the limited number of sampled cubs may have influenced the analysis of risk factors for Cryptosporidium infection. A comparison with similar surveys in other carnivore hosts would be welcome.

Reply: We recognize that since clinical infection by Cryptosporidium may be age-related, it would be interesting to assess infection in a larger sample size of cub individuals in future studies. However, we would like to point out that all of our samples were formed without evidence of diarrhoea, suggesting that the infections were asymptomatic, as indicated in current lines 233 and 234.

  1. A limited number of minor inaccuracies is highlighted in the attached file.

Reply: Corrected in current Table 2 and lines 332 and 428.